# Ambient air pollutant mixture and lung function among children in Fresno, California

Wenxin Lu[1,2*], Ellen A. Eisen[1], Liza Lutzker[3], Elizabeth Noth[1], Tim Tyner[4], Fred Lurmann[5], S. Katharine Hammond[1], Stephanie Holm[6‡], John R. Balmes[1,7‡]

1 Division of Environmental Health Sciences, School of Public Health, University of California Berkeley, Berkeley, California, United States of America, 2 The Herbert Wertheim School of Public Health and Human Longevity Science, University of California San Diego, La Jolla, California, United States of America, 3 Division of Epidemiology, School of Public Health, University of California Berkeley, Berkeley, California, United States of America, 4 Central California Asthma Collaborative, Fresno, California, United States of America, 5 Sonoma Technology, Petaluma, California, United States of America, 6 Stephanie Holm Consulting, Vancouver, British Columbia, Canada, 7 Department of Medicine, University of California San Francisco, San Francisco, California, United States of America

‡ Co-senior authors.
* wluac@berkeley.edu

## Abstract

### Background

Ambient air pollutants such as particulate matter (PM), ozone ($O_3$), and nitrogen dioxide ($NO_2$) have been associated with lower lung function among children. However, the reported associations could be due to correlation with other pollutants.

### Objective

We investigate the relationships between exposures to eight ambient air pollutants and children's lung function and apply mixture analysis to identify key contributors to health effects.

### Methods

The Children's Health and Air Pollution Study (CHAPS) in Fresno, California, is a prospective cohort study that recruited 299 children and assessed their lung function at two visits, at approximately 7 and 9 years of age. The children's forced expiratory volume in the first second ($FEV_1$), forced vital capacity (FVC), and $FEV_1$/FVC ratio were standardized using the Global Lung Function Initiative (GLI) race-neutral calculators. We assessed the children's average daily residential exposures to $PM_{2.5}$, $PM_{10}$, nitrogen oxides ($NO_x$), $NO_2$, $O_3$, carbon monoxide (CO), elemental carbon (EC), and polycyclic aromatic hydrocarbons (PAHs), during the 1-week, 1-month, 3-month, 6-month, and 12-month periods before each visit, and the 2 years between visits. We applied linear mixed-effect models and quantile-based g-computation (q-gcomp) for statistical analysis.

**Data availability statement:** The de-identified analysis dataset and R code are available on the Open Science Framework platform: https://doi.org/10.17605/OSF.IO/8TXBQ.

**Funding:** This research was supported by the Children's Health and Air Pollution Study (CHAPS), an NIH / EPA-funded Children's Environmental Health Research Center (EPA: RD83543501, NIH: ES022849), and two additional grants (NIEHS R24 ES03088, NIH 5T32HD101364).

**Competing interests:** The authors have declared that no competing interests exist.

**Abbreviations:** BKMR, Bayesian Kernel Machine Regression; CHAPS, Children's Health and Air Pollution Study; CI, Confidence interval; CO, Carbon monoxide; DAG, Directed acyclic graph; EC, Elemental Carbon; FEV1, Forced expiratory volume in the first second; FVC, Forced vital capacity; IPCW, Inverse probability of censoring weights; IQR, Interquartile range; L, Liter; $NO_2$, Nitrogen dioxide; $NO_x$, Nitrogen oxides; $O_3$, Ozone; PAH, Polycyclic aromatic hydrocarbon; PAH456, 4–5- and 6-ring polycyclic aromatic hydrocarbon compounds; PM, Particulate matter; $PM_{2.5/10}$, Particulate matter with an aerodynamic diameter of 2.5/10 μm or less; q-gcomp, Quantile-based g-computation; SD, Standard deviation; SES, Socioeconomic status; VOC, Volatile organic compound

## Results

The children's exposures to the eight ambient air pollutants exhibited high intercorrelation: Seven air pollutants were positively correlated, while $O_3$ exposures were negatively correlated with the other pollutants. Higher $PM_{10}$ was associated with lower $FEV_1$ and $FEV_1/FVC$ ratio, and the associations were strongest for the 3-month exposure timeframe. Q-gcomp also identified $PM_{10}$ as the key pollutant associated with lower $FEV_1$ and $FEV_1/FVC$ ratio.

## Conclusion

Among the eight ambient air pollutants, $PM_{10}$ was the strongest risk factor for impaired lung function among children in Fresno. Ambient air pollution levels in this community exceed regulatory standards and are harmful to children's lung function.

## Introduction

Ambient air pollution is one of the most prevalent environmental exposures and has been linked with various adverse health effects. Children are the most vulnerable to the health effects of air pollution because their biological systems are still developing, and they inhale more air per bodyweight compared to adults [1]. The growing respiratory systems of children are directly impacted by air pollution exposure, which leads to respiratory diseases and impaired lung function [2–4]. As summarized by Garcia et al. in 2021, both short-term and long-term exposure to ambient air pollution have been found to affect lung function among children [3]. Exposures to ambient air pollutants, such as particulate matter with aerodynamic diameters of 2.5 and 10 μm or less ($PM_{2.5}$ and $PM_{10}$, respectively), ozone ($O_3$), carbon monoxide (CO) elemental carbon (EC), nitrogen dioxide ($NO_2$), and polycyclic aromatic hydrocarbons (PAHs), have been associated with lower lung function in asthmatic and non-asthmatic children [5–15].

Several knowledge gaps on air pollution and child lung function remain [3], including understanding whether the observed associations of $NO_2$ and child lung function are causal or due to correlations with other pollutants or pollutant mixtures, such as traffic-related air pollution. The spatial-temporal distributions of ambient air pollutants are often correlated due to common emission sources and photochemical reactions [16–18]. Traditional analyses that investigate air pollutants individually and sequentially fail to differentiate between causal effects and associations due to confounding by correlated air pollutants. Mixture analysis methods, such as weighted quantile sum regression, quantile-based g-computation (q-gcomp), and Bayesian Kernel Machine Regression (BKMR), can address correlated multi-pollutant mixtures, estimate the total mixture effect, and compare the relative importance of different pollutants on the outcome of interest [19–22].

Two recent studies have investigated ambient air pollution exposure and child lung function using mixture analysis methods. A study in Boston applied BKMR to investigate prenatal exposures to $NO_2$, $O_3$, and PM constituents on child lung function at 7

years old, and found that $O_3$, organic carbon, and ammonium were most strongly associated with reduced lung function [23]. A study in Fresno applied BKMR and found that the combined exposures to $NO_2$, $O_3$, PM, and pesticides were associated with slightly lower lung function among asthmatic children, although the findings were not statistically significant [24]. No study has investigated mixtures of air pollutants beyond the most commonly studied air pollutants--$NO_2$, $O_3$, and PM--or applied other mixture analysis methods.

In this study, we investigate the relationships between exposures to eight ambient air pollutants, namely $PM_{2.5}$, $PM_{10}$, $NO_2$, $NO_x$, CO, $O_3$, EC, and 4–5- and 6-ring PAH compounds (PAH456), and lung function in a cohort of 7–9-year-old children in central California. We first conduct traditional single-pollutant analyses to estimate the individual effect of each pollutant, then apply q-gcomp to estimate the joint effect of air pollutant mixture and compare the relative contributions of each pollutant. We also investigate the relationship at different exposure timeframes to identify key exposure timeframes that are more influential on children's lung functions.

## Materials and methods

### Study design and population

The Children's Health and Air Pollution Study (CHAPS) is a prospective cohort that recruited 299 children in the Fresno Unified School District and followed them for 2 years. Details of the recruitment process and eligibility criteria were described elsewhere [25]. We invited the eligible 6–8-year-old children and their parents or guardians for a first visit at the study center between May 2015 and May 2017, and a second visit between June 2017 and April 2019. Written informed permission was obtained from each accompanying parent or guardian. All recruitments and study center visits were completed between May 1, 2015, and April 30, 2019. The two visits were scheduled at least 21 months apart for all participants. Among the 299 children who completed the first visit, 218 (73%) returned for the second visit. All study protocols were approved by the Institutional Review Boards at the University of California Berkeley, Berkeley, California, United States of America.

### Exposure assessment

We collected hourly ambient concentrations for $PM_{2.5}$, $PM_{10}$, $NO_2$, $NO_x$, CO, and $O_3$ from the United States Environmental Protection Agency Air Quality System monitoring sites and the Fresno central monitoring station. Hourly ambient EC and PAH456 concentrations were measured with aethalometers (model AE42, Magee Scientific, Berkeley, CA) and photoelectric aerosol sensors (model PAS2000, EcoChem Analytics, League City, TX), respectively. We developed spatial-temporal models for the eight air pollutants using inverse distance-squared weighting or linear regression with mixed effects, depending on the pollutant. More details on exposure monitoring and spatial-temporal modeling can be found in previous publications. [26–28]

We obtained and geocoded the complete residential histories of the participants at the two study center visits and calculated the participants' daily average residential exposures to each pollutant using the spatial-temporal models. We further calculated the participants' average residential exposures to the eight pollutants during the 1-week, 1-month, 3-month, 6-month, and 12-month periods before each visit. At all exposure assessment steps, observations with less than 75% data completeness were treated as missing.

### Outcome assessment

At both visits, the study children were asked to perform up to eight attempts of spirometry tests to obtain three spirometry curves of high quality. Spirometry was performed by trained study staff according to American Thoracic Society/European Respiratory Society (ATS/ERS) standards using an Easy One spirometer (ndd Medical Technologies; Zurich, Switzerland) [29]. Under physician supervision, trained spirometry graders analyzed the three best curves selected by the spirometer's algorithm and evaluated the acceptability of the curves. The acceptability criteria for $FEV_1$ and FVC are summarized in

Table 7 of the ATS/ERS technical statement [29]. For each child at each visit, we recorded their best forced expiratory volume in the first second ($FEV_1$), forced vital capacity (FVC), and $FEV_1$/FVC ratio among their acceptable spirometry measurements. We then standardized the $FEV_1$, FVC, and $FEV_1$/FVC ratio measurements as z-scores based on the age and sex of the children, using the Global Lung Function Initiative (GLI) race-neutral spirometry calculators [30].

## Covariates

The participants' health and sociodemographic characteristics, including age, sex, height, and weight at visits, asthma ever-diagnosis, race/ethnicity, household income, home ownership, parental education, and household second-hand smoke exposure, were collected from questionnaires administered at each visit. We also recorded the season of each visit as winter (November to February), spring (March to June), or autumn (July to October), considering the local climate in Fresno, California. We obtained the neighborhood socioeconomic status (SES) from the CalEnviroScreen (version 3.0) 5-year estimates (2011–2015) of census-tract level education, unemployment, and poverty [31]. We matched the neighborhood SES variables to the geocoded residential addresses and calculated averages weighted by the number of days the participant lived at each address during exposure periods of interest.

We constructed a directed acyclic graph (DAG) a priori, illustrating the pathways between ambient air pollution exposure and children's lung function (S1 File). Based on this DAG, we identified confounders and mediators and selected a sufficient adjustment set needed to block all confounding pathways. The sufficient adjustment set of confounders consists of season (categorical), neighborhood SES (education, unemployment, and poverty rates as linear continuous variables), race/ethnicity (categorical), household income (binary, >30k or <30k US dollars), and house ownership (binary). All statistical analyses described below were adjusted for variables in the sufficient adjustment set.

## Statistical analysis

We examined the relationship between average air pollution exposures during the 1-week, 1-month, 3-month, 6-month, and 12-month periods before each visit and the children's standardized lung function outcomes at each visit. We conducted single-pollutant analysis using linear mixed-effect models and mixture analysis using q-gcomp with cluster-based bootstrapping, which accounts for the repeated measurements from the same participants. Q-gcomp is a mixture analysis method that estimates the expected change in the outcome if all pollutant exposures increase by one quantile. It can also produce weights that indicate the relative contributions of each of the pollutants to the outcome of interest. The mathematical details and implementation procedures of q-gcomp have been described by Keil et al [20]. In the q-gcomp models, we divided exposures into four quartiles, specified a Monte Carlo sample size of 1000, and used 500 bootstrap replicates to estimate confidence intervals (CI).

In all analyses described above, we applied inverse probability of censoring weights (IPCW) to observations at the second visit to minimize the risk of selection bias due to potential differential loss to follow-up [32,33]. A similar IPCW has been applied in a previous analysis of the CHAPS cohort [34]. A DAG illustrating the selection bias causal pathways is presented in S2 File, with a detailed explanation of IPCW calculation and how it reduces the risk of selection bias is included in its footnotes. Briefly, we assigned weights calculated as the inverse of the conditional probability of completing the second visit, given the participants' exposure and lung function at the first visit. Applying this IPCW is equivalent to creating a pseudo-population where censorship status is independent of past exposures and outcomes, breaking the causal pathways for selection bias.

We also conducted three sets of sensitivity analyses. First, we repeated the main analyses while restricting to participants who were able to perform at least one, two, or three spirometry blows with acceptable $FEV_1$ measures based on the ATS/ERS acceptability criteria [29]. Lung function measurements are less accurate among children because it is more difficult for children to perform voluntary breathing maneuvers of acceptable quality compared to adults [35]. By restricting to children who were able to perform high-quality spirometry tests repeatedly, we expect more accurate outcome

assessment and less bias due to measurement error. However, spirometry test failures are also associated with respiratory diseases and symptoms [36], and restricting to more rigid repeatability criteria may increase the risk of selection bias [37]. We compared the results from the analyses with different repeatability criteria and evaluated the risks of the different sources of bias.

Second, we repeated the q-gcomp analysis with the seven pollutants except for $O_3$. This is because exposure to $O_3$ was negatively correlated with the exposures to the other pollutants (this will be described later in the results section). Q-gcomp estimates the expected change in outcome under the hypothetical treatment of increasing all pollutant exposure levels. Thus, including $O_3$ in the exposure set may reduce the statistical power and increase the risk of violating the positivity assumption, i.e., there must be a non-zero probability of receiving the treatment for all combinations of observed characteristics. We compared the results from the q-gcomp analyses with seven pollutants (no $O_3$) and eight pollutants (with $O_3$) to examine the role of $O_3$ and evaluate the risk of positivity violation. Finally, we repeated the analyses without IPCW weights to test the robustness of the results.

All statistical analyses were conducted with the software R version 4.4.0. To avoid erroneous inference caused by multiple testing in both single-pollutant analysis and mixture analysis, patterns of point estimates and CIs will be interpreted instead of individual p-values or statistical significance. The manuscript was prepared using the Strengthening the Reporting of Observational Studies in Epidemiology (STROBE) checklist for cohort studies [38].

## Results

**Table 1** summarizes the baseline sociodemographic characteristics of the CHAPS study participants. The participants were 47% female, 80% Hispanic/Latinx, and on average 7.5 years old at the first visit. Most participants were from low SES households that had annual incomes less than $30,000 (66%), did not own homes (78%), and had child health insurance covered by Medicaid (85%). Participating children from households that did not own homes and whose mothers were unemployed were less likely to have completed the second visit at 9 years old.

The distributions of the participating children's lung function measurements are illustrated in **Fig 1**. The participants' standardized $FEV_1$, FVC, and $FEV_1$/FVC ratios were approximately normally distributed, with the average z-scores for $FEV_1$ (mean = −0.25, SD = 1.24) and $FEV_1$/FVC ratio (mean = −0.28, SD = 1.15) slightly below zero, indicating higher risks of airway obstruction compared to the GLI international reference population. This is consistent with the high prevalence of asthma (21%) at baseline among the participants (**Table 1**). The participants' average $FEV_1$ and FVC increased by 0.36 L and 0.47 L between the first and second visits, while the average z-scores for $FEV_1$ and FVC increased by 0.13 and 0.25, indicating that the participants' lung function growth between 7 and 9 years old was, on average, higher than the GLI international reference population.

The participants' average exposures to the eight ambient air pollutants before the two visits are summarized in **Fig 2**. All participants were exposed to 12-month average $PM_{2.5}$ levels higher than the US National Ambient Air Quality Standard (NAAQS) primary standard of 9 μg/m³ [39] and 12-month average $PM_{10}$ levels higher than the California Ambient Air Quality Standard of 20 μg/m³ [40]. Compared to the first visit (2015−2017), exposures to $O_3$ decreased, while exposures to CO, $NO_2$, $PM_{10}$, and PAH456 increased before the second visit (2017−2019). The exposures to the eight ambient air pollutants were correlated across all timeframes (**S3 File**): seven pollutants, except for $O_3$, were positively correlated (r range: 0.20–0.97), while $O_3$ was negatively correlated with all other pollutants (r range: −0.83–0.13). The negative correlations between $O_3$ and other pollutants were likely due to its photochemical conversions with $NO_2$ [17,41], as well as the different seasonal patterns between primary air pollutants and $O_3$ [42].

The associations between the average pollutant exposures before each visit and the children's standardized lung function measurements estimated from single-pollutant models are illustrated in **Fig 3** (for $FEV_1$ z-scores), **Fig 4** (for FVC z-scores), and **Fig 5** (for $FEV_1$/FVC ratio z-scores). The results of the sensitivity analysis that were restricted to children who were able to perform at least one, two, or three good spirometry blows are also presented in **Figs 3-5**.

**Table 1. Baseline characteristics of the Children's Health and Air Pollution Study (CHAPS) participants, summarized by whether the participant completed both visits.**

| | Completed both visits (N = 218) | Completed the first visit only (N = 81) | All participants (N = 299) |
|---|---|---|---|
| **Sex** | | | |
| Female | 102 (46.8%) | 38 (46.9%) | 140 (46.8%) |
| **Age at first visit** | | | |
| Mean (SD) | 7.5 (0.6) | 7.5 (0.6) | 7.5 (0.6) |
| **Race/ethnicity** | | | |
| Hispanic/Latinx | 178 (81.7%) | 60 (74.1%) | 238 (79.6%) |
| **Home owning** | | | |
| Yes | 57 (26.1%) | 8 (9.9%) | 65 (21.7%) |
| **Annual household income** | | | |
| <15k USD | 58 (26.6%) | 25 (30.9%) | 83 (27.8%) |
| 15-30k USD | 85 (39.0%) | 28 (34.6%) | 113 (37.8%) |
| **Maternal education** | | | |
| <High school | 71 (32.6%) | 24 (29.6%) | 95 (31.8%) |
| High school/GED | 106 (48.6%) | 38 (46.9%) | 144 (48.2%) |
| College and advanced | 41 (18.8%) | 18 (22.2%) | 59 (19.7%) |
| **Maternal employment** | | | |
| Employed | 98 (45.0%) | 26 (32.1%) | 124 (41.5%) |
| **Child's Health insurance** | | | |
| Covered by work | 32 (14.7%) | 10 (12.3%) | 42 (14.0%) |
| Covered by government | 184 (84.4%) | 69 (85.2%) | 253 (84.6%) |
| **Asthma status** | | | |
| Ever-diagnosis of asthma | 48 (22.0%) | 15 (18.5%) | 63 (21.1%) |

We compared the demographic and health characteristics of children who were able to perform different numbers of spirometry blows and found that the proportion of asthmatic children was higher among those who performed three good spirometry tests (30%) or zero good spirometry tests (26%), compared to those who performed one (16%) or two (15%) good spirometry tests. The differences in the proportion of asthmatic children were statistically significant (p = 0.03).

Higher exposures to ambient $PM_{10}$ were associated with lower $FEV_1$ z-score, and the association was the strongest at the 3-month exposure timeframe before each visit (**Fig 3**, panel 7). Increasing average $PM_{10}$ exposure over the past 3 months from the 25th to the 75th percentile was associated with a 0.28 (95% CI: 0.01–0.55) lower $FEV_1$ z-score. The association between $PM_{10}$ exposure and $FEV_1$ was consistent under different repeatability criteria. A similar pattern was found for the associations between average $PM_{10}$ exposures and FEV1/FVC ratio z-score, especially after setting a more rigid repeatability criterion of performing at least three good blows (**Fig 5**, panel 7): Among children who were able to perform at least three good spirometry blows, increasing average $PM_{10}$ exposure over the past 3 months from the 25th to the 75th percentile was associated with a 0.47 (95% CI: 0.08, 0.86) lower $FEV_1$/FVC z-score. Higher 12-month average CO exposures were also associated with lower $FEV_1$/FVC ratio z-scores (**Fig 5**, panel 1). We observed an unexpected positive association between 6-month average PAH456 exposure and FEV1 z-score, but the association did not persist under more rigid repeatability criteria or other exposure time frames (**Fig 3**, panel 6). No association was found between air pollution exposure and FVC z-scores at the two visits (**Fig 4**).

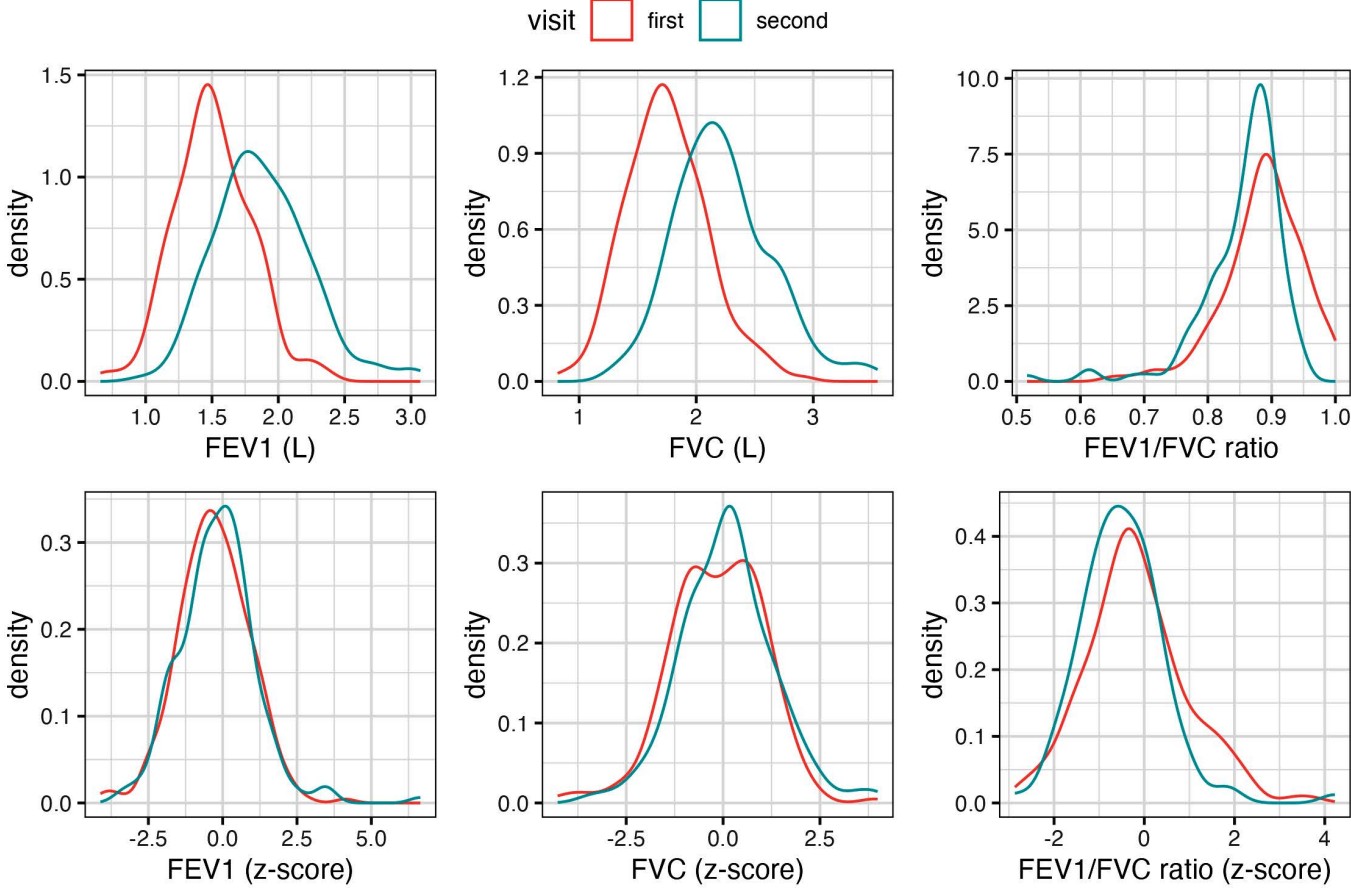

**Fig 1. Density distribution of the raw and standardized lung function measurements of the Children's Health and Air Pollution Study (CHAPS) participants at the first (7-year-old) and the second (9-year-old) visits.**

The q-gcomp mixture analysis results between exposures to the mixture of the eight ambient air pollutants and lung function measurements are summarized in S1 Fig. Increasing the 12-month average exposures to all eight pollutants by one quartile is associated with a slightly lower $FEV_1$/FVC ratio z-score (estimate = −0.18, 95% CI: −0.41–0.05), which was consistent under different repeatability criteria. No association was found for other combinations of exposure time frame and lung function measurements. The q-gcomp weights that represent the relative contributions of pollutants are summarized in S2 Fig (for $FEV_1$ z-scores), S3 Fig (for FVC z-score), and S4 Fig (for $FEV_1$/FVC ratio z-score). The q-gcomp models identified different key pollutants at different exposure time frames, with $PM_{10}$ exposure contributing to lower $FEV_1$ and $FEV_1$/FVC z-scores at all time frames. This is also consistent with the results from the single-pollutant models (**Fig 3**, **Fig 5**). In addition, the q-gcomp weights for PAH456 were almost consistently positive in models for FEV1 and FEV1/FVC ratio (S2 Fig, S4 Fig) and are consistent with the slightly positive association between PAH456 and FEV1z-score in the single-pollutant analysis.

We conducted several sensitivity analyses to test the robustness of the results and minimize the risk of bias. First, repeating the analyses without IPCW yielded similar results with no qualitative difference. Second, we conducted a sensitivity analysis that included the seven pollutants except for $O_3$. Compared with the analysis with eight pollutants (S1 Fig), restricting to seven pollutants yielded similar estimates with narrower CIs across all models (S5 Fig).

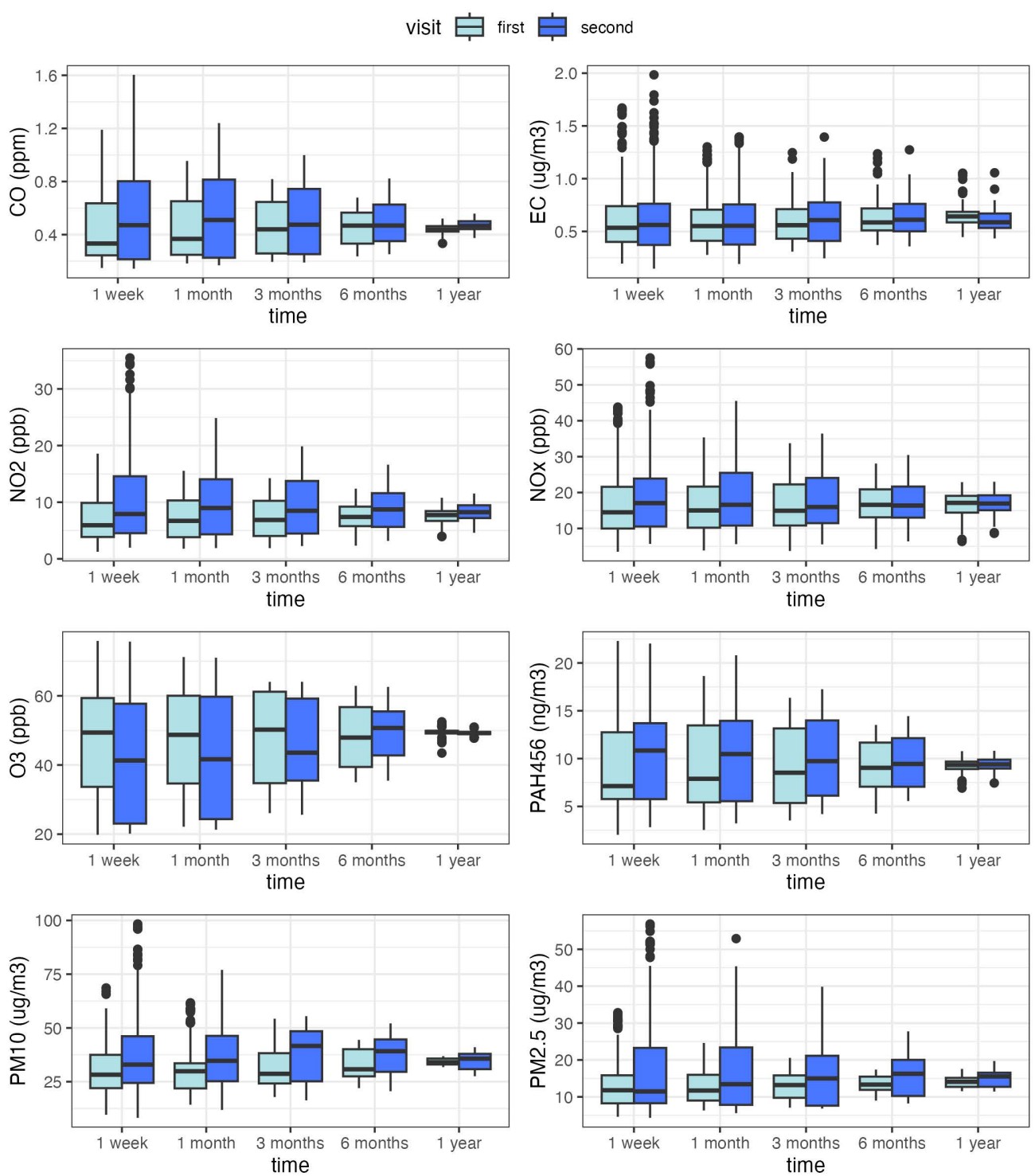

**Fig 2. Distributions of the 1-week, 1-month, 3-month, 6-month, and 12-month average residential air pollutant exposures before the first (7-year-old) and the second (9-year-old) visits among Children's Health and Air Pollution Study (CHAPS) participants.** Note: data completeness was 88%−92% for PAH456 exposures before the first visit. For all other pollutants and exposures before the second visit, data completeness was at least 96%.

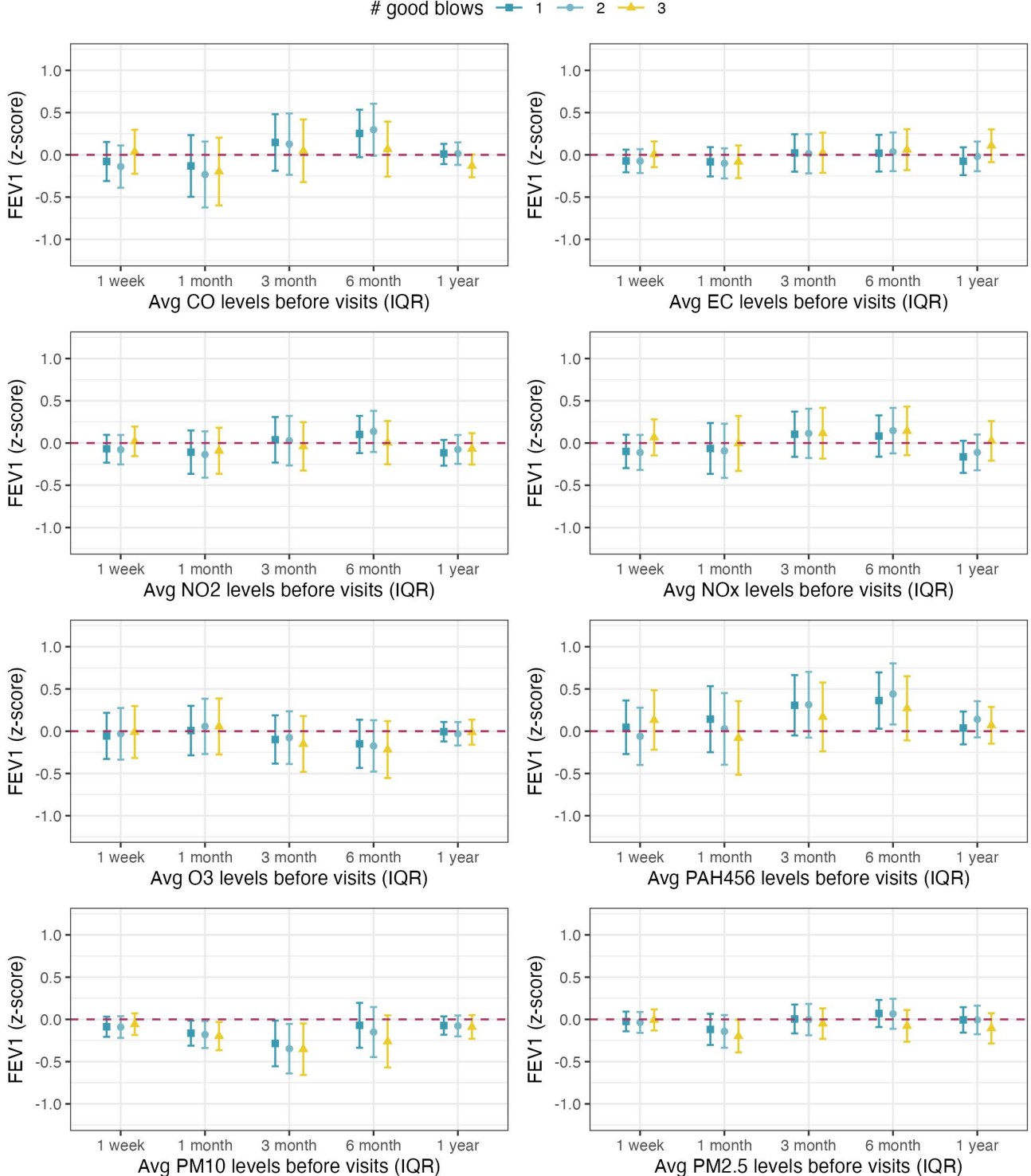

**Fig 3. Associations between 1-week, 1-month, 3-month, 6-month, and 12-month average exposures to eight ambient air pollutants and the standardized FEV1 measurements among Children's Health and Air Pollution Study (CHAPS) participants.** Note: All models were adjusted for the sufficient adjustment set (season, neighborhood SES, race/ethnicity as a proxy for structural racism, and household SES) and applied IPCW in linear mixed-effect models. Setting repeatability criteria of at least 1, 2, or 3 good spirometry blows restricted the analyses to 454, 384, and 214 observations, respectively.

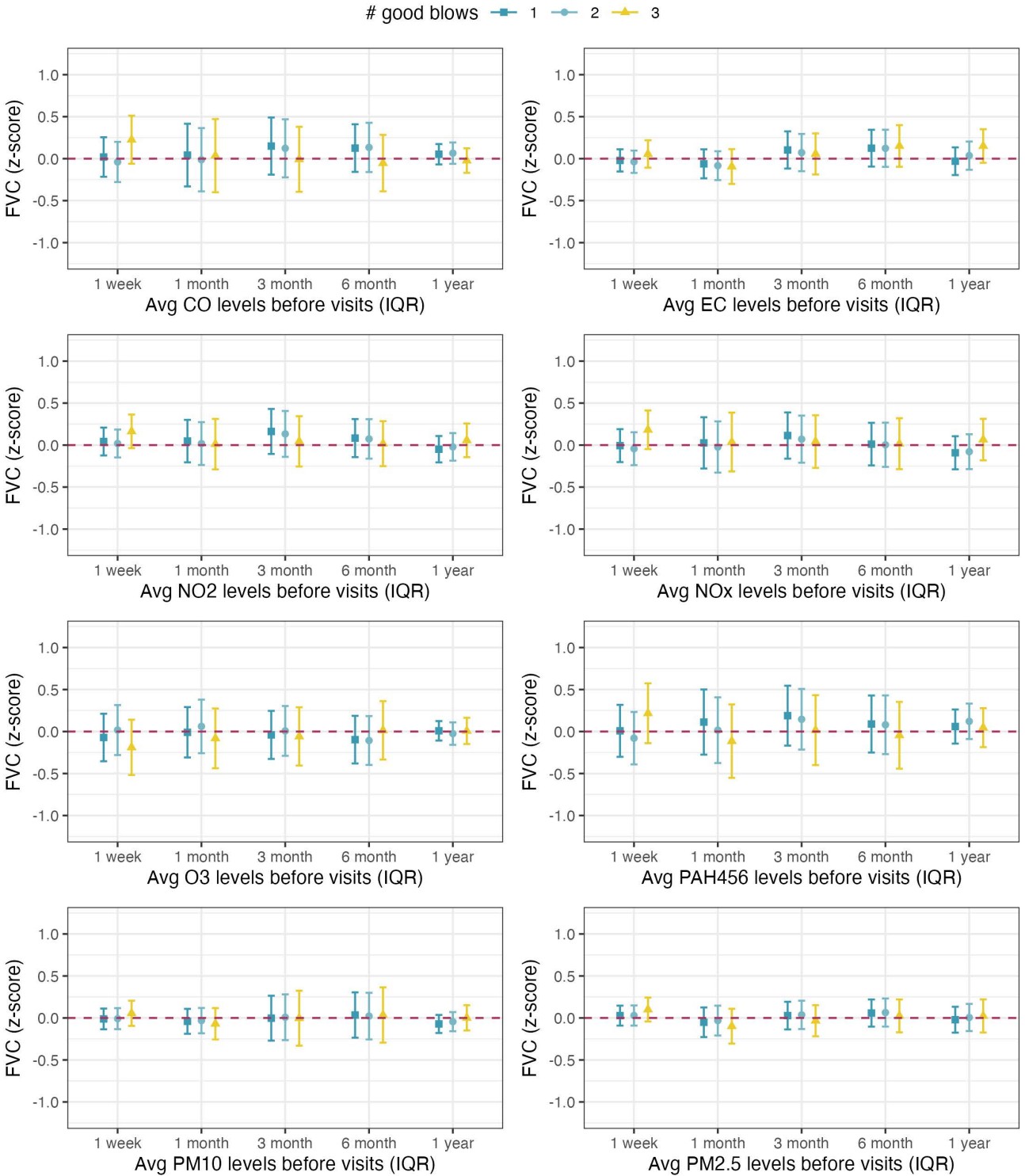

**Fig 4. Associations between 1-week, 1-month, 3-month, 6-month, and 12-month average exposures to eight ambient air pollutants and the standardized FVC measurements among Children's Health and Air Pollution Study (CHAPS) participants.** Note: All models were adjusted for the sufficient adjustment set (season, neighborhood SES, race/ethnicity as a proxy for structural racism, and household SES) and applied IPCW in linear mixed-effect models. Setting repeatability criteria of at least 1, 2, or 3 good spirometry blows restricted the analyses to 454, 384, and 214 observations, respectively.

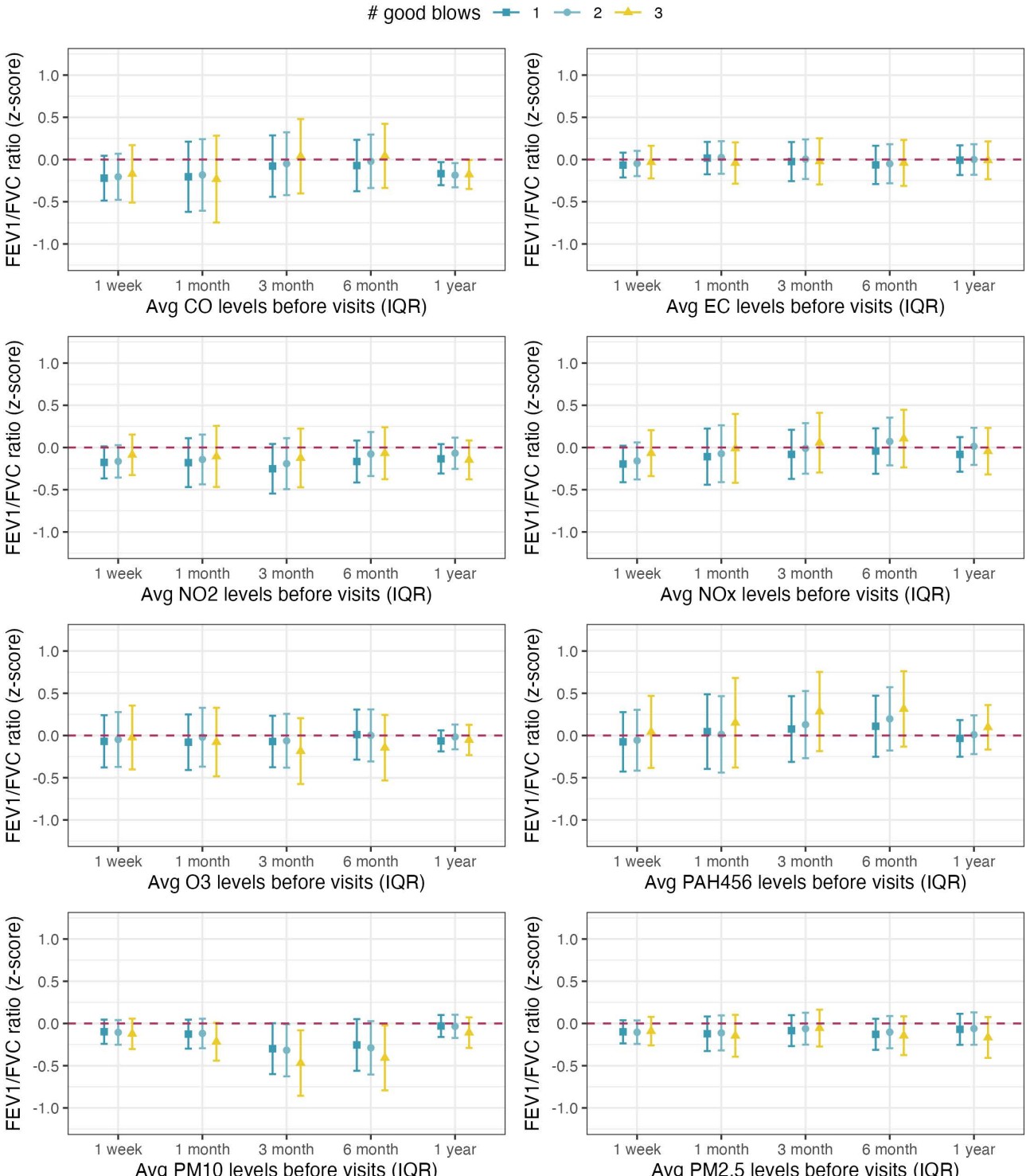

**Fig 5. Associations between 1-week, 1-month, 3-month, 6-month, and 12-month average exposures to eight ambient air pollutants and the standardized FEV1/FVC ratio measurements among Children's Health and Air Pollution Study (CHAPS) participants.** Note: All models were adjusted for the sufficient adjustment set (season, neighborhood SES, race/ethnicity as a proxy for structural racism, and household SES) and applied IPCW in linear mixed-effect models. Setting repeatability criteria of at least 1, 2, or 3 good spirometry blows restricted the analyses to 454, 384, and 214 observations, respectively.

## Discussion

We studied a cohort of 299 children recruited from a primarily Hispanic/Latinx population in Fresno, California. The study children were exposed to high levels of particulate air pollution above the national and state air quality standards and had a high prevalence of asthma at baseline. We found that higher $PM_{10}$ exposures before visits, especially during the 3 months before visits, were associated with lower standardized $FEV_1$ and $FEV_1$/FVC ratio, suggesting a potential effect of dust on respiratory obstruction. Q-gcomp mixture analysis also identified $PM_{10}$ as the key pollutant contributing to poorer lung function; however, no significant association was found between exposure to the pollutant mixture and lung function.

$PM_{10}$ contains dust particles with various chemical compositions that airway defense mechanisms (e.g., mucus layer and macrophage phagocytosis) protect against when inhaled. Exposure to high levels of dust may exceed the capacity of the respiratory defenses, disrupt the epithelium, and induce airway inflammation [43]. Several existing studies have investigated $PM_{10}$ exposure and child lung function at various time frames. A Swedish study found that $PM_{10}$ exposure during the first year of life was associated with lower $FEV_1$ at 8 years old [44]. An English study found that $PM_{10}$ exposure on the previous day was associated with a small reduction in lung function [45]. No study has investigated medium-term (weeks to months) $PM_{10}$ exposure and child lung function and compared the relationship at different exposure time frames. Our finding that $PM_{10}$ exposure was associated with lower FEV1 and FEV1/FVC ratio, with the strongest association found for 3-month exposure, is complementary to the existing literature. $PM_{10}$ is also an identified risk factor for chronic obstructive pulmonary disease [46,47], which is consistent with our finding that $PM_{10}$ is associated with lower $FEV_1$ and $FEV_1$/FVC ratio, which indicate airflow obstruction.

The strongest associations between $PM_{10}$ and $FEV_1$ and $FEV_1$/FVC ratio were found for the 3-month exposure before visits, which might be due to two reasons. First, dust exposure at moderate levels, as observed in this study, may take time to accumulate and disrupt the airway defense mechanisms, induce airflow obstruction, and reduce $FEV_1$ levels. This may be why $PM_{10}$ exposures during the past 3 months showed stronger associations with $FEV_1$ and $FEV_1$/FVC ratio compared to $PM_{10}$ exposures during the past week or month. Second, exposures averaged over longer periods have smaller data variations (**Fig 2**), leading to reduced statistical power, which might be the reason for the diminishing associations from 3 months to 6 months and 12 months exposure periods.

We conducted sensitivity analyses under different repeatability criteria by including children who were able to perform at least one, two, or three good spirometry blows at each visit. We found that asthmatic children were more likely to perform either zero or at least three good spirometry tests. This bimodal distribution may be due to two factors: First, it is harder for asthmatic patients with more severe disease and lower lung function to perform satisfactory spirometry tests [36]. Second, some asthmatic children may be more familiar with spirometry testing due to their medical treatment history. In addition, including all children who could perform one good spirometry blow resulted in larger measurement errors, especially for FVC, which requires strong efforts to exhale their full expiratory capacity, and is hard for young children to perform [35]. Therefore, results under less rigid repeatability criteria are subject to a bias towards the null due to random measurement errors, as well as an upward bias caused by excluding more asthmatic children with poorer lung function. Results under more rigid repeatability criteria may also be subject to a slight downward bias due to restricting to more asthmatic children. Overall, results under different repeatability criteria should be evaluated together. We only interpreted results that are consistent under different repeatability criteria to minimize the risks of the biases specified above. In addition, restricting our analyses to the three best curves selected by the device algorithm may have also contributed to potential outcome misclassification, both upward and downward bias.

We observed an unexpected positive association between 6-month PAH456 and FEV1 z-score in the single-pollutant analysis. Although this association was not consistently positive across different repeatability criteria, q-gcomp weights also suggested a positive contribution of PAH456 to FEV1. The positive association may be due to the negative correlation between 6-month PAH456 exposure and O3, which was slightly negatively associated with FEV1 (S3 File). This positive association should be interpreted with caution because the narrow exposure range for PAH456 was very narrow

in our study (IQR = 4.9ng/m3; Note: The Occupational Safety and Health Administration's 8-hr permissible exposure limit is 0.2 mg/m3). Future studies with wider PAH exposure ranges are needed to assess the effects of PAHs on lung function growth.

Q-gcomp addresses the intercorrelation among multiple exposures by employing weighted quantile sum regression and estimates the mixture effect through simulation under various scenarios [20]. We selected q-gcomp over other mixture analysis methods for three reasons: First, q-gcomp results are easy to interpret. Second, unlike WQS, q-gcomp does not require the directional homogeneity assumption, i.e., the associations between all exposures and the outcome are in the same direction or null, which is likely violated in this study due to the negative correlation between $O_3$ and other pollutants (S3 File). Third, the moderate sample size and narrow exposure range for several pollutants, including CO, $NO_2$, and PAH456, may not be sufficient to support highly flexible non-parametric methods like BKMR. In our application of q-gcomp, we observed low statistical power that could be due to the loss of data variation at the categorization step. In our sensitivity analysis with seven pollutants excluding $O_3$ (S5 Fig), we observed much narrower CIs compared to the main analysis with eight pollutants (S1 Fig). Estimating the hypothetical effect of increasing the levels for all mixture components may not be statistically efficient if the natural distributions for certain components are negatively correlated with others. In the q-gcomp analyses, we identified a similar set of key pollutants as the single-pollutant analyses. However, the pollutant weights produced by q-gcomp (S2–S4 Figs) are less stable compared to the point estimates and CIs in single-pollutant analysis (Figs 3–5), which might also be due to the loss of data variation during exposure categorization.

This study collected exposure data for eight ambient air pollutants with high temporal resolution and applied a novel mixture analysis method that accounts for the high correlation among pollutants. We conducted repeated spirometry tests with rigorous quality assessment by experienced physicians and performed sensitivity analyses with different repeatability criteria. We applied the new GLI race-neutral calculator published in 2023 to avoid the marginalization of disadvantaged communities, which is a historical problem in pulmonary research and clinical practice [30,48,49]. Leveraging available data, we minimized the risk of potential biases by constructing a DAG *a priori* for confounder identification and applying IPCW to account for the loss-to-follow-up at the second visit.

This study has a few limitations. The residential air pollution exposures were extrapolated from air pollutant monitors. Although this is a common practice in air pollution epidemiology studies, the spatial extrapolation and the fact that residential exposure does not capture the children's exposures inside the homes or at school compromised the accuracy of exposure assessment. The potential exposure measurement error is non-differential regarding the children's lung function status and is expected to cause a bias towards the null. Other ambient air pollutants, such as VOCs and sulfur dioxide, could also affect pulmonary function [50,51] but were not measured in this cohort. Finally, the moderate sample size and relatively low exposures to air pollutants other than PM in this population (**Fig 2**) might be the reasons for the null associations between most air pollutants and child lung function. Future studies with larger sample sizes in populations with higher variations in air pollutant levels should be conducted to further explore the relationship between ambient air pollutant mixtures and lung function.

## Supporting information

**S1 Fig. Quantile-based g-computation (q-gcomp) results for the associations between 1-week, 1-month, 3-month, 6-month, and 12-month average exposures to the mixture of eight ambient air pollutants and the standardized lung function measurements.** Note: All models adjusted for the sufficient adjustment set (season, neighborhood SES, race/ethnicity as a proxy for structural racism, and household SES) and applied IPCW. Cluster-based bootstrapping was used to account for repeated measures. Setting repeatability criteria of at least 1, 2, or 3 good spirometry blows at both visits restricted the analyses to 454, 384, and 214 observations, respectively.
(TIFF)

**S2 Fig. Quantile-based g-computation (q-gcomp) pollutant contribution weights for the associations between 1-week, 1-month, 3-month, 6-month, and 12-month average exposures to the mixture of eight ambient air pollutants and the standardized forced expiratory volume in the first second (FEV1).** Note: The point estimates and confidence intervals of the same q-gcomp models are summarized in Figure S2. Pollutant contribution weights for the same model (same exposure time frame and repeatability criteria) are directly comparable. Negative weights represent negative contributions (harmful) to the lung function outcome.
(TIFF)

**S3 Fig. Quantile-based g-computation (q-gcomp) pollutant contribution weights for the associations between 1-week, 1-month, 3-month, 6-month, and 12-month average exposures to the mixture of eight ambient air pollutants and the standardized forced vital capacity (FVC).** Note: The point estimates and confidence intervals of the same q-gcomp models are summarized in Figure S2. Pollutant contribution weights for the same model (same exposure time frame and repeatability criteria) are directly comparable. Negative weights represent negative contributions (harmful) to the lung function outcome.
(TIFF)

**S4 Fig. Quantile-based g-computation (q-gcomp) pollutant contribution weights for the associations between 1-week, 1-month, 3-month, 6-month, and 12-month average exposures to the mixture of eight ambient air pollutants and the standardized FEV1/FVC ratio.** Note: The point estimates and confidence intervals of the same q-gcomp models are summarized in Figure S2. Pollutant contribution weights for the same model (same exposure time frame and repeatability criteria) are directly comparable. Negative weights represent negative contributions (harmful) to the lung function outcome.
(TIFF)

**S5 Fig. Quantile-based g-computation (q-gcomp) results for the associations between 1-week, 1-month, 3-month, 6-month, and 12-month average exposures to the mixture of seven ambient air pollutants and the standardized lung function measurements.** Note: All models adjusted for the sufficient adjustment set (season, neighborhood SES, race/ethnicity as a proxy for structural racism, and household SES) and applied IPCW. Cluster-based bootstrapping was used to account for repeated measures. Setting repeatability criteria of at least 1, 2, or 3 good spirometry blows at both visits restricted the analyses to 454, 384, and 214 observations, respectively. Seven ambient air pollutants, except for ozone, are analyzed. The results for the same analyses with eight air pollutants, including ozone, are shown in Figure S2.
(TIFF)

**S1 File. Directed acyclic graph characterizing the causal pathways between ambient air pollution and children's lung function.**
(PDF)

**S2 File. The causal pathways of selection bias due to differential censoring at the second visit in the Children's Health and Air Pollution Study (CHAPS) study.**
(PDF)

**S3 File. Correlation plots for the Children's Health and Air Pollution Study (CHAPS) participants' average residential exposures to eight ambient air pollutants before visits.**
(PDF)

## Author contributions

**Conceptualization:** Stephanie Holm, John R Balmes.

**Data curation:** Liza Lutzker, Elizabeth Noth, Fred Lurmann, S Katharine Hammond.

**Formal analysis:** Wenxin Lu.

**Funding acquisition:** Stephanie Holm, John R Balmes.

**Investigation:** Elizabeth Noth, Tim Tyner, Fred Lurmann, S Katharine Hammond.

**Methodology:** Ellen A Eisen, Fred Lurmann, Stephanie Holm.

**Project administration:** Liza Lutzker, Tim Tyner.

**Resources:** John R Balmes.

**Supervision:** Ellen A Eisen, Tim Tyner, Stephanie Holm, John R Balmes.

**Validation:** Elizabeth Noth, Fred Lurmann.

**Visualization:** Wenxin Lu.

**Writing – original draft:** Wenxin Lu.

**Writing – review & editing:** Ellen A Eisen, Tim Tyner, S Katharine Hammond, Stephanie Holm, John R Balmes.

## Acknowledgments

The authors would like to thank the UCSF-Fresno research team (Griselda Aguilar, Christian Bonilla, Karina Corona, Cynthia Cortez, Alexa Lopez, Carolina Orozco, and Janna Blaauw) for their hard work in conducting the clinical visits, undergraduate research assistants (Barune Thapa, Natalie Myren, Kimberly Meyer, and Peter Buto for their contributions to data entry and geocoding, Beth MacDonald for her contributions to data management, and De'Asia Thomas and Alexandra Tien-Smith for contributing to spirometry grading. We deeply appreciate the contributions of all participants in providing valuable data for this research.

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
