## [Decision Letter · Decision Letter 0]

23 Sep 2025

Dear Dr. Lu,

Thank you for submitting your manuscript to PLOS ONE. After careful consideration, we feel that it has merit but does not fully meet PLOS ONE’s publication criteria as it currently stands. Therefore, we invite you to submit a revised version of the manuscript that addresses the points raised during the review process.

We look forward to receiving your revised manuscript.

Kind regards,

LS Katrina Li

Academic Editor

PLOS ONE

Journal Requirements:

“This research was supported by the Children's Health and Air Pollution Study (CHAPS), an NIH / EPA-funded Children’s Environmental Health Research Center (EPA: RD83543501, NIH: ES022849), and two additional grants (NIEHS R24 ES03088, NIH 5T32HD101364).”

3. In the online submission form, you indicated that “The datasets generated and analyzed during the current study are not publicly available to protect participant information and privacy, but are available from the corresponding author on reasonable request.”

4. Please upload a copy of Figure 6, to which you refer in your text on page 21. If the figure is no longer to be included as part of the submission please remove all reference to it within the text.

Reviewers' comments:

Reviewer's Responses to Questions

**Comments to the Author**

1. Is the manuscript technically sound, and do the data support the conclusions?

Reviewer #1: Yes

Reviewer #2: Yes

2. Has the statistical analysis been performed appropriately and rigorously?

Reviewer #1: Yes

Reviewer #2: I Don't Know

3. Have the authors made all data underlying the findings in their manuscript fully available?

Reviewer #1: Yes

Reviewer #2: No

4. Is the manuscript presented in an intelligible fashion and written in standard English?

Reviewer #1: Yes

Reviewer #2: Yes

Reviewer #1: This is a novel and well-conceived study that makes a valuable contribution to the field. To strengthen the manuscript for publication, I recommend improving figure clarity, presenting results with and without IPCW to assess their impact, and providing more details on the q-gcomp and spatiotemporal model parameters for reproducibility. Additionally, sharing the R scripts and related data in open repositories (e.g., GitHub) would enhance transparency and reproducibility.

Reviewer #2: This is a well written manuscript that contributes to important public health questions regarding commonly encountered air pollutants and child lung function impacts.

1. Modeling strategy – Many exposure metrics are examined in both the individual pollutant models and the mixtures analysis. This appears exploratory in that no a priori hypothesis is presented regarding how these were selected and interpretation of biological plausibility comes only in interpretation of findings (emphasizing the adverse effect of the 3 mo averaging period for PM10). Quantile base g comp is used for the mixture analysis.

Provide a rationale for selection of Q g comp. The authors could have considered BKMR-DLM given there was no a prior hypothesis or biological conceptual model about averaging periods.

2. The authors correctly note the multiple testing nature of the analyses and focus on trends. As described, “To avoid erroneous inference caused by multiple testing in the single-pollutant analysis, patterns of point estimates and confidence intervals (CIs) will be interpreted instead of individual p-values or statistical significance.”

The mixture analysis (5 exposure periods) could also be considered as multiple testing and should be included in the statement above.

In addition, given this approach to interpretation, the authors should report and discuss the relatively consistent “positive” effect observed for PAH (figure S5, S7, Figure 3,4,5).

3. Spirometry interpretation description could be improved. It is not entirely clear how grading of spirometry influenced inclusion in analyses. In the outcome assessment section, the authors state: For each child at each visit, we recorded their best forced expiratory volume in the first second (FEV1), forced vital capacity (FVC), and FEV1/FVC ratio among their graded spirometry curves. Please explain what grading implies here. Were some dropped because graded as unacceptable? What criteria were used to determine grading. Similarly, some additional detail regarding sensitivity analyses based on spirometry performance would be helpful. The paper describes sensitivity analysis based on including participants who could perform at least one, two, or three acceptable spirometry blows with valid FEV1 measures. How was acceptable determined/validity?

4. Given the difference in performance for spirometry for children with and without asthma and a reasonable suspicion that the effects of these pollutants and exposure periods may vary for children with vs. without asthma, did the authors consider analysis by asthma status?

5. The Abstract states: “Ambient air pollutants such as particulate matter (PM) and nitrogen dioxide (NO2) “ – Add ozone here given it is among the most consistently associated with effect on lung function in children in relation to both shorter and longer term exposure.

6. The introduction refers to mixture models. “Mixture analysis methods, such as weighted quantile sum regression and quantile-based g-computation (q-gcomp), can address correlated multi-pollutant mixtures. BKMR should be noted here as well, particularly given the next examples from the literature employ BKMR.

7. The results state: “The exposures to the eight ambient air pollutants were highly correlated across all timeframes.” While the pollutants were clearly correlated there was a range of degree of correlation from more modest (0.1-0.2) to “highly”. I suggest removing the word highly here.

**Do you want your identity to be public for this peer review?** For information about this choice, including consent withdrawal, please see our Privacy Policy

Reviewer #1: **Yes: ** Mohammad Fayaz

Reviewer #2: No

---

## [Author Response · Author response to Decision Letter 1]

13 Oct 2025

We deeply appreciate all the constructive feedback from the reviewers and editors. These comments and suggestions have helped us improve the quality and clarity of this manuscript. See below for responses to individual comments. All line numbers reference the manuscript document with tracked changes.

Response to reviewers’ comments:

Reviewer #1: This is a novel and well-conceived study that makes a valuable contribution to the field. To strengthen the manuscript for publication, I recommend improving figure clarity, presenting results with and without IPCW to assess their impact, and providing more details on the q-gcomp and spatiotemporal model parameters for reproducibility. Additionally, sharing the R scripts and related data in open repositories (e.g., GitHub) would enhance transparency and reproducibility.

This manuscript investigates the relationships between exposures to eight ambient air pollutants (PM2.5, PM10, NO2, NOx, CO, O3, EC, and PAH456) and lung function among 299 children aged 7–9 years in Fresno, California, a disadvantaged community with high ambient air pollution. Using both traditional single-pollutant linear mixed-effect models and quantile-based g-computation (q-gcomp) for mixture analysis, the study identifies PM10 as the key contributor associated with lower standardized FEV1 and FEV1/FVC ratio, particularly for 3-month average exposures.

The study is novel in applying a mixture analysis method that accounts for correlations among multiple pollutants and evaluates different exposure timeframes, which extends prior research that focused on a limited set of pollutants or single-pollutant models. Strengths include high-quality spirometry with repeatability criteria, the application of GLI race-neutral calculators, rigorous adjustment for confounders via a DAG, and sensitivity analyses using inverse probability of censoring weights (IPCW). Weaknesses include some figure clarity issues, limited reporting of model parameters for q-gcomp, and reliance on residential exposure estimates rather than personal monitoring. Overall, this is a well-conceived and valuable study that could be suitable for publication after addressing the following points.

Major issues (must be addressed for the manuscript to proceed)

• Figure clarity: Many figures (Figures 1–5, S3–S8) have blurry axis labels and tick marks. Improving resolution and label readability is necessary.

Thank you for your suggestion. We believe that the compromised resolution in those figures is due to the compression during the PDF exportation process. For all figures in the main manuscript and supplements, we have uploaded the original figures with high resolution. For the supplementary files, we have replaced the original PDF file with an updated version that minimizes compression. If the editors and production team deem it necessary, we are also happy to reproduce the figures to further improve the resolution.

Model details: Provide more explicit details about q-gcomp model parameters, including the number of quantiles, bootstrapping settings, and handling of O3 correlations, to allow reproducibility by other researchers.

Thank you for this suggestion. We have added a clarifying sentence (lines 223-225): “In the q-gcomp models, we divided exposures into four quartiles, specified a Monte Carlo sample size of 1000, and used 500 bootstrap replicates to estimate confidence intervals (CI). ” In terms of the negative correlation between ozone and other pollutants, we conducted a sensitivity analysis using the other seven pollutants, compared to the main analysis with all eight pollutants. This sensitivity analysis has been described in lines 262-270.

Minor issues (suggested improvements)

• Code and data sharing: Make the R scripts and related datasets available in an open repository (e.g., GitHub), if possible, to enhance transparency and reproducibility.

Thank you for this suggestion. We have uploaded a de-identified analysis dataset, R code to the Open Science Framework. The files can be viewed at https://osf.io/8txbq/?view_only=60214516a7954ce69a63bc4572b967e1. The project is currently private, and we will make it public once the manuscript has been officially accepted for publication.

• Figures and tables: Consider combining or simplifying supplemental figures (S5–S8) for clarity and ease of interpretation.

Thank you for this suggestion. We have rearranged the supplementary materials, and Figures S5-S8 are now S3-S6. We plotted Figures S3 – S5 in the supplement to follow the same layout as Figures 3-5 in the main manuscript, and Figure S6 to follow the same format as Figure S2. These corresponding Figure formats help with comparison and interpretation and are currently the best way we could think of to present the weight and estimate information from the q-gcomp analysis. We are open to any specific suggestions on how to improve figure clarity.

• Results validation: Include analyses with and without IPCW (or discuss explicitly if already done) to show the robustness of the findings to potential selection bias.

Thank you for this suggestion. We have conducted the same set of analyses with no IPCW. All the results are very similar to the main results with IPCW. For example, the association between 3-month PM10 exposure and FEV1 z-score among participants with at least one good spirometry curve was -0.28 (-0.55, -0.01) with IPCW and -0.29 (-0.56, -0.02) without IPCW. We did not include the analysis results without IPCW because, as you have pointed out in the previous comment, the supplementary materials are already quite dense in information. We modified the manuscript to describe the analyses with no IPCW as a sensitivity analysis as follows:

Lines 270-271: Finally, we repeated the analyses without IPCW weights to test the robustness of the results.

Lines 436-438: We conducted several sensitivity analyses to test the robustness of the results and minimize the risk of bias. Repeating the analyses without IPCW yielded similar results with no qualitative difference.

• Text clarity: Minor language and formatting improvements could enhance readability, particularly in the Results and Discussion sections.

Thank you for this suggestion. We have modified the language in several sections to improve clarity. We also screened for grammatical errors in the results and discussion sections.

Reviewer #2: This is a well written manuscript that contributes to important public health questions regarding commonly encountered air pollutants and child lung function impacts.

1. Modeling strategy – Many exposure metrics are examined in both the individual pollutant models and the mixtures analysis. This appears exploratory in that no a priori hypothesis is presented regarding how these were selected and interpretation of biological plausibility comes only in interpretation of findings (emphasizing the adverse effect of the 3 mo averaging period for PM10). Quantile base g comp is used for the mixture analysis.

Provide a rationale for selection of Q g comp. The authors could have considered BKMR-DLM given there was no a prior hypothesis or biological conceptual model about averaging periods.

Thank you for this comment. You are correct that our analysis does not have any a priori hypothesis or biological conceptual model. The main reasons that we chose q-gcomp instead of BKMR-DLM are: (1) Our moderate sample size and narrow exposure range for several pollutants (especially CO, NO2, and PAH456) are not sufficient to support highly flexible non-parametric methods like BKMR-DLM; (2) We aimed for a simple interpretation from the mixture analysis for the non-statistical audience, including physicians, policy makers, and the general public; (3) For BKMR-DLM or DLM in general, it is ideal to have exposure assessments for multiple equally spaced time periods for each participant-visit. However, exposure assessments for additional time frames and related data quality assurance were not feasible within the scope of this project.

We also added some rationale for the mixture analysis method selection in the discussion section. The sentences (lines 533-539) read: “We selected q-gcomp over other mixture analysis methods for three reasons: First, q-gcomp results are easy to interpret. Second, unlike WQS, q-gcomp does not require the directional homogeneity assumption, i.e., the associations between all exposures and the outcome are in the same direction or null, which is likely violated in this study due to the negative correlation between O3 and other pollutants (S2 File). Third, the moderate sample size and narrow exposure range for several pollutants, including CO, NO2, and PAH456, may not be sufficient to support highly flexible non-parametric methods like BKMR.”

2. The authors correctly note the multiple testing nature of the analyses and focus on trends. As described, “To avoid erroneous inference caused by multiple testing in the single-pollutant analysis, patterns of point estimates and confidence intervals (CIs) will be interpreted instead of individual p-values or statistical significance.”

The mixture analysis (5 exposure periods) could also be considered as multiple testing and should be included in the statement above.

In addition, given this approach to interpretation, the authors should report and discuss the relatively consistent “positive” effect observed for PAH (figure S5, S7, Figure 3,4,5).

Thank you for this suggestion. We have added mixture analysis in the multiple testing statement. The sentence now reads: “To avoid erroneous inference caused by multiple testing in both single-pollutant analysis and mixture analysis, patterns of point estimates and CIs will be interpreted instead of individual p-values or statistical significance.”

We also added some description and interpretation of the slightly positive association between PAH and FEV1. The added sentences are:

Results (lines 410 – 413): We observed an unexpected positive association between 6-month average PAH456 exposure and FEV1 z-score, but the association did not persist under more rigid repeatability criteria or other exposure time frames (Fig 3, panel 6).

Results (lines 432 – 435): In addition, the q-gcomp weights for PAH456 were almost consistently positive in models for FEV1 and FEV1/FVC ratio (S3 Fig, S5 Fig) and are consistent with the slightly positive association between PAH456 and FEV1z-score in the single-pollutant analysis.

Discussion (lines 521 – 530): We observed an unexpected positive association between 6-month PAH456 and FEV1 z-score in the single-pollutant analysis. Although this association was not consistently positive across different repeatability criteria, q-gcomp weights also suggested a positive contribution of PAH456 to FEV1. The positive association may be due to the negative correlation between 6-month PAH456 exposure and O3, which was slightly negatively associated with FEV1 (Fig 3). This positive association should be interpreted with caution because the narrow exposure range for PAH456 was very narrow in our study (IQR = 4.9ng/m3; Note: The Occupational Safety and Health Administration’s 8-hr permissible exposure limit is 0.2mg/m3). Future studies with wider PAH exposure ranges are needed to assess the effects of PAHs on lung function growth.

3. Spirometry interpretation description could be improved. It is not entirely clear how grading of spirometry influenced inclusion in analyses. In the outcome assessment section, the authors state: For each child at each visit, we recorded their best forced expiratory volume in the first second (FEV1), forced vital capacity (FVC), and FEV1/FVC ratio among their graded spirometry curves. Please explain what grading implies here. Were some dropped because graded as unacceptable? What criteria were used to determine grading. Similarly, some additional detail regarding sensitivity analyses based on spirometry performance would be helpful. The paper describes sensitivity analysis based on including participants who could perform at least one, two, or three acceptable spirometry blows with valid FEV1 measures. How was acceptable determined/validity?

Thank you for this question. The acceptability of the spirometry measurements was determined based on the ATS/ERS technical statement, which is currently the gold standard for spirometry. The statement can be found at: https://www.atsjournals.org/doi/10.1164/rccm.201908-1590ST, and the repeatability criterion required for FEV1 and FVC can be found in Table 7. We also added more information on spirometry grading and related sensitivity analysis in the methods section. The modified sentences read:

Lines 173-184: Under physician supervision, trained spirometry graders analyzed the three best curves selected by the spirometer’s algorithm and evaluated the acceptability of the curves. The acceptability criteria for FEV1 and FVC are summarized in Table 7 of the ATS/ERS technical statement (29). For each child at each visit, we recorded their best forced expiratory volume in the first second (FEV1), forced vital capacity (FVC), and FEV1/FVC ratio among their acceptable spirometry measurements.

Lines 243-246: . First, we repeated the main analyses while restricting to participants who were able to perform at least one, two, or three spirometry blows with acceptable FEV1 measures based on the ATS/ERS acceptability criteria (29).

4. Given the difference in performance for spirometry for children with and without asthma and a reasonable suspicion that the effects of these pollutants and exposure periods may vary for children with vs. without asthma, did the authors consider analysis by asthma status?

Thank you for this question. Stratifying the analysis by asthma status is an interesting research question worth exploring. In our cohort, there are only 56 and 40 children with asthma who had valid FEV1 values at the first and the second visit, respectively. This very limited sample size will result in compromised statistical power in the stratified analysis, with lower precision and even wider confidence intervals compared to the main analysis.

5. The Abstract states: “Ambient air pollutants such as particulate matter (PM) and nitrogen dioxide (NO2) “ – Add ozone here given it is among the most consistently associated with effect on lung function in children in relation to both shorter and longer term exposure.

Thank you for this suggestion. We have added ozone in the first sentence of the abstract as suggested.

6. The introduction refers to mixture models. “Mixture analysis methods, such as weighted quantile sum regression and quantile-based g-computation (q-gcomp), can address correlated multi-pollutant mixtures. BKMR should be noted here as well, particularly given the next examples from the literature employ BKMR.

Thank you for this suggestion. We have added BKMR in that sentence as suggested.

7. The results state: “The exposures to the eight ambient air pollutants were highly correlated across all timeframes.” While the pollutants were clearly correlated there was a range of degree of correlation from more modest (0.1-0.2) to “highly”. I suggest removing the word highly here.

Thank you for this suggestion. We have removed the word “highly” in that sentence as suggested.

Responses to journal requirements:

We have modified our manuscript and files based on these style requirements.

“This research was supported by the Children's Health and Air Pollution Study (CHAPS), an NIH / EPA-funded Children’s Environmental Health Research Center (EPA: RD83543501, NIH: ES022849), and two additional grants (NIEHS R24 ES03088, NIH 5T32HD101364).”

Please state what role the funders took in the study. If the funders had no role, please state: "The funders had no role in study design, data collection and analysis, decision to publish, or preparation of the manus

---

## [Editor Report · Decision Letter 1]

15 Oct 2025

Ambient air pollutant mixture and lung function among children in Fresno, California

PONE-D-25-43765R1

Dear Dr. Lu,

We’re pleased to inform you that your manuscript has been judged scientifically suitable for publication and will be formally accepted for publication once it meets all outstanding technical requirements.

Kind regards,

LS Katrina Li

Academic Editor

PLOS ONE

---

## [Editor Report · Acceptance letter]

PONE-D-25-43765R1

PLOS ONE

Dear Dr. Lu,

I'm pleased to inform you that your manuscript has been deemed suitable for publication in PLOS ONE. Congratulations! Your manuscript is now being handed over to our production team.

Kind regards,

on behalf of

Dr. LS Katrina Li

Academic Editor

PLOS ONE